# Impact of Baseline Anti-ABO Antibody Titer on Biliary Complications in ABO-Incompatible Living-Donor Liver Transplantation

**DOI:** 10.3390/jcm13164789

**Published:** 2024-08-14

**Authors:** Se-Hyeon Yu, Hye-Sung Jo, Young-Dong Yu, Pyoung-Jae Park, Hyung-Joon Han, Sang-Jin Kim, Syahrul Hadi Kamarulzaman, Dong-Sik Kim

**Affiliations:** 1Division of HBP Surgery and Liver Transplantation, Department of Surgery, Korea University Anam Hospital, Korea University College of Medicine, Seoul 02841, Republic of Korea; vict1216@naver.com (S.-H.Y.); hust1351@naver.com (Y.-D.Y.); kimds1@korea.ac.kr (D.-S.K.); 2Division of Transplantation and Vascular Surgery, Department of Surgery, Korea University Guro Hospital, Korea University College of Medicine, Seoul 02841, Republic of Korea; pyoungjae@hanmail.net; 3Division of Hepatobiliopancreas and Transplant Surgery, Korea University Ansan Hospital, Ansan-si 05505, Republic of Korea; hjhan@korea.ac.kr (H.-J.H.); khjginigini@hanmail.net (S.-J.K.)

**Keywords:** ABO-incompatible living donor liver transplantation, biliary complication

## Abstract

**Background:** Although advancements in desensitization protocols have led to increased ABO-incompatible (ABOi) living-donor liver transplantation (LDLT), a higher biliary complication rate remains a problem. This study evaluated the effect of baseline anti-ABO antibody titers before desensitization on biliary complications after ABOi LDLT. **Methods:** The study cohort comprised 116 patients in the ABO-compatible group (ABOc), 29 in the ABOi with the low titer (<1:128) group (ABOi-L), and 14 in the high titer (≥1:128) group (ABOi-H). **Results:** Biliary complications occurred more frequently in the ABOi-H group than in the ABOi-L and ABOc groups (7 [50.0%] vs. 8 [27.6%] and 24 [20.7%], respectively, *p* = 0.041). Biliary complication-free survival was significantly worse in the ABOi-H group than in the other groups (*p* = 0.043). Diffuse intrahepatic biliary strictures occurred more frequently in the ABOi-H group than in the other groups (*p* = 0.005). Multivariable analysis revealed that the high anti-ABO antibody titer (≥1:128) is an independent risk factor for biliary complications (hazard ratio 3.943 [1.635–9.506]; *p* = 0.002). **Conclusions:** A high baseline anti-ABO antibody titer (≥1:128), female sex, and hepatic artery complications are significant risk factors for biliary complications.

## 1. Introduction

Living-donor liver transplantation (LDLT) has reduced waitlist mortality and mitigated the scarcity of deceased donors in many countries [1]. The adoption of desensitization protocols, which include rituximab and total plasma exchange (TPE), has enabled ABO-incompatible (ABOi) LDLT to emerge as a crucial alternative when ABO-compatible (ABOc) donors are limited [2,3]. However, a significantly higher rate of biliary complications, including diffuse intrahepatic biliary strictures (DIHBSs), is an Achilles heel of ABOi LDLT [4,5,6,7]. Despite advances in immunosuppressive strategies, the etiology and risk factors for biliary complications following ABOi LDLT remain elusive.

The most prevalent biliary complications after LDLT include biliary strictures and leakage, which can cause recurrent cholangitis, severe peritonitis, and septic shock. Managing these complications usually requires frequent intervention, including endoscopic retrograde cholangiography, percutaneous transhepatic cholangiography, and surgical revision [8]. Furthermore, biliary complications can impede graft function recovery and may result in fatal graft loss, particularly during the early postoperative period. Even when effectively resolved, repeated invasive procedures extend hospital stays and significantly increase medical costs.

Although the cause of biliary complications in ABOi LDLT has not yet been fully elucidated, they are known to be caused by immunological factors that target the biliary epithelium [9,10]. Despite a reduction in the anti-ABO antibody titer immediately before transplantation through the desensitization protocol, the incidence of biliary complications remains higher in ABOi LDLT [9], suggesting a possible association with baseline anti-ABO antibody titers. Therefore, this study aimed to evaluate the impact of baseline anti-ABO antibody titers before desensitization on biliary complications and to identify the risk factors for biliary complications following ABOi LDLT.

## 2. Materials and Methods

### 2.1. Study Population

This study included consecutive patients who underwent LDLT between 2009 and 2022. Patients aged <18 years were excluded. There were 116 and 43 patients in the ABOc LDLT and ABOi LDLT groups, respectively (Figure 1). Patients with ABOi LDLT were stratified into two cohorts based on their baseline anti-ABO antibody titers, either equal and above or below 1:128. Finally, the study population was divided into the following three groups: ABOc (n = 116), ABOi with low-baseline anti-ABO antibody titer (ABOi-L) (n = 29; <1:128), and ABOi with high baseline anti-ABO antibody titer (ABOi-H) (n = 14; ≥1:128). A prospectively maintained database of all donors and recipients was retrospectively reviewed.

### 2.2. Desensitization Protocols and Immunosuppressants

Baseline anti-ABO antibody titers were evaluated in patients scheduled for ABOi LDLT. Recipients were admitted to the hospital two weeks before surgery and administered a single dose of rituximab (300 mg/m^2^/BSA). TPE was initiated one week before transplantation, and anti-ABO antibody titers were monitored the day after TPE. The anti-ABO antibody titer goal immediately before transplantation was ≤1:8. The number of TPE sessions was determined based on the degree of titer reduction. The anti-ABO antibody titer was monitored daily until postoperative day (POD) 7 and subsequently monitored thereafter.

During surgery, patients received 20 mg of basiliximab (Simulect^®^, Novartis Pharmaceuticals, UK Ltd., London, UK) and 500 mg of methylprednisolone. Following liver transplantation, a triple regimen, including a calcineurin inhibitor, mycophenolate mofetil, and steroids, was initiated, depending on the recipient’s clinical condition. Tacrolimus was administered at a targeted blood concentration of 8–10 ng/mL, and mycophenolate mofetil was used at 1.0 g/day from POD 1 in all recipients. Methylprednisolone was gradually tapered, changed to prednisolone after POD 7, and withdrawn by POD 100. 

### 2.3. Surgical Procedure

Detailed surgical procedures for both donors and recipients have been described previously [11]. During donor surgery, bile duct division was performed using intraoperative cholangiography or indocyanine green fluorescence imaging in open and laparoscopic procedures, respectively. In this study, duct-to-duct anastomosis with an internal stent was a standardized surgical procedure for bile duct reconstruction. In contrast, Roux-en-Y hepaticojejunostomy (RYHJ) was performed for patients who presented with primary sclerosing cholangitis and a history of previous bilio-digestive operations. Furthermore, the precise radiation field was verified in patients who underwent liver-directed radiation therapy. RYHJ was performed if the liver hilum was affected by a high-dose radiation field. Bile duct anastomosis was performed using polydioxanone sutures, 6-0, intermittently. The RYHJ was performed in a retrocolic and isoperistaltic manner.

### 2.4. Diagnosis and Definition of Biliary Complication

During the immediate postoperative period, when bile leakage was suspected in abdominal drainage, computed tomography scanning was performed to check for fluid collection. In addition, when an elevation in serum bilirubin levels was accompanied by increased alkaline phosphatase and gamma-glutamyl transferase levels without any immunological or other special causes, computed tomography scans were conducted to confirm the suspected biliary stricture. Once the intrahepatic bile duct was dilated on imaging, endoscopic retrograde cholangiography (ERC) was performed for accurate diagnosis and treatment. Patients who underwent RYHJ were diagnosed and treated with percutaneous transhepatic cholangiography (PTC) because of their inability to access the biliary tract using ERC. In this study, biliary complications were defined as biliary leakage or strictures confirmed using ERC or PTC. Bile flow disturbances caused by an internal tube placed during duct anastomosis were not classified as biliary complications.

### 2.5. Statistical Analysis

Statistical analyses were conducted to compare and identify differences in baseline characteristics, operative variables, biliary complication-free survival, and risk factors for biliary complications among the three groups. Biliary complication-free survival rates were determined using the Kaplan–Meier method, and comparisons were made using the log-rank test. Other variables were compared using the chi-square test, Fisher’s exact test, the Mann–Whitney *U* test, the Kruskal–Wallis test, and a one-way analysis of variance. Cox proportional hazard regression analysis was used to determine the risk factors for biliary complications. A multivariable analysis was performed on factors with *p*-values ≤ 0.1 in the univariable analysis (left graft, multiple graft bile duct opening, female sex, ABO-incompatibility with high anti-ABO antibody titer [≥1:128], hepatocellular carcinoma, warm ischemic time [≥20 min], and hepatic artery complication). Statistical significance was defined as a *p*-value < 0.05. All statistical analyses were performed using IBM SPSS Statistics for Windows (version 22.0; IBM Corp., Armonk, NY, USA).

## 3. Results

### 3.1. Baseline Characteristics of the Recipients and Donors

The baseline characteristics of the recipients and donors in the ABOc, ABOi-L, and ABOi-H groups are shown in Table 1. The median baseline anti-ABO antibody titer and number of TPE sessions performed before transplantation were significantly higher in the ABOi-H group than in the ABOi-L group (median 256 [range 128–1024] vs. 32 [range 8–64], *p* < 0.001 and 4 [2–11] vs. 2 [0–4], *p* < 0.001, respectively). Additionally, the frequency of autoimmune liver disease was significantly different among the three groups (*p* = 0.022). The model for end-stage liver disease and Child–Pugh scores were comparable among the ABOc, ABOi-L, and ABOi-H groups (*p* = 0.288 and *p* = 0.632, respectively).

Regarding donors, grafts with macrosteatosis (≥20%) were comparable among the three groups (*p* = 0.416). The laparoscopic approach was performed similarly in all three groups (*p* = 0.131). The graft-to-recipient weight ratio was higher in the ABOi-H group than in the ABOc and ABOi-L groups with borderline significance (1.05 [0.73–1.32] vs. 0.98 [0.60–1.89] and 0.94 [0.70–1.94], *p* = 0.052).

### 3.2. Operative Findings and Postoperative Outcomes

Regarding the characteristics of the bile duct, most grafts had a single bile duct opening (80 [60.9%] in the ABOc group vs. 21 [72.4%] in the ABOi-L group vs. 10 [71.4%] in the ABOi-H group, *p* = 1.000) (Table 2). The bile duct size was comparable between the groups (minimum size, 4 [1–13] vs. 4 [2–8] vs. 5 [2–10], *p* = 0.909). Ductoplasty for multiple bile duct openings was comparable between the groups (*p* = 0.921). For bile duct reconstruction, most patients underwent duct-to-duct anastomosis with an internal stent as the standard procedure (101 [87.1%] vs. 27 [93.1%] vs. 14 [100.0%], *p* = 0.803).

Major complication (Clavien–Dindo grade ≥ IIIa) rates were comparable between the ABOc, ABOi-L, and ABOi-H groups (58 [50.0%] vs. 14 [48.3%] vs. 9 [64.3%], *p* = 0.571) (Table 2). However, the incidence of biliary complications differed significantly between the groups (24 [20.7%] vs. 8 [27.6%] vs. 7 [50.0%], *p* = 0.041). However, no significant differences were observed for the 30-day and 90-day mortality rates among the three groups (*p* = 1.000 and *p* = 1.000, respectively).

### 3.3. Biliary Complications

The incidence of biliary strictures was significantly higher in the ABOi-H group than in the ABOc and ABOi-L groups (7 [50.0%] vs. 24 [20.7%] and 8 [27.6%], respectively; *p* = 0.041) (Figure 2). Regarding the type of biliary complication, most patients developed a biliary stricture across the groups (*p* = 0.317). However, DIHBS occurred more frequently in the ABOi-H group than in the ABOc and ABOi-L groups (3 [21.4%] vs. 2 [1.7%] and 0 [0%], *p* = 0.005). Two patients with DIHBS underwent re-transplantation due to an unresolved biliary stricture (one in the ABOc group and one in the ABOi-H group). 

The treatment period for biliary complications using ERC or PTC was significantly longer in the ABOi-L group, and there was no significant difference between the ABOc and ABOi-H groups (748 days [113–1546] vs. 280 [1–643] and 118 [5–416], respectively; *p* = 0.001). Among the patients with biliary complications who required treatment for >6 months, 54% (13/24) were in the ABOc group, 62% (5/8) were in the ABOi-L group, and 28% (2/7) were in the ABOi-H group. No significant difference existed among the three groups for the rate of percutaneous transhepatic bile drainage implementation during the treatment period (45.8%, 37.5%, and 42.9%, respectively, *p* = 1.000). None of the patients experienced major post-ERCP complications affecting the clinical course, such as bleeding, pancreatitis, or gastrointestinal perforation.

### 3.4. Risk Factors for Biliary Complication-Free Survival

The biliary complication-free survival (BCFS) rates at 1, 3, and 6 months in the ABOc group were 94.8%, 87.7%, and 83.3%, respectively. The corresponding rates in the ABOi-L and ABOi-H groups were 93.1%, 89.7%, and 86.1% and 85.7%, 78.6%, and 71.4%, respectively (Figure 3). The BCFS rate was significantly different among the three groups, with the lowest rate observed in the ABOi-H group (*p* = 0.043).

Multivariable Cox proportional hazard regression analysis of BCFS revealed that a high baseline anti-ABO antibody titer (≥1:128) is an independent risk factor (hazard ratio [HR], 3.493 [1.635–9.506]; *p* = 0.002) (Table 3). Other significant factors include recipient female sex (3.307 [1.271–8.604], *p* = 0.014) and postoperative hepatic artery complications (3.505 [1.313–9.354], *p* = 0.012). Three of the four patients who developed hepatic artery complications had biliary complications.

## 4. Discussion

This study aimed to identify the factors contributing to biliary complications in ABOi LDLT. No consensus exists regarding the levels of anti-ABO antibody titers considered safe for antibody-mediated rejection or associated risk factors for complications [10,12]. The two primary biliary complications after LDLT are bile leakage and biliary strictures. Recently, an international multicenter study of 3633 cases reported that important risk factors associated with bile leaks included multiple bile duct anastomoses, RYHJ, and a history of major abdominal surgery. The risk factors for biliary strictures included blood loss exceeding 1 L, previous abdominal surgery, and ABO incompatibility [9].

After confirming biliary complications according to the baseline anti-ABO antibody titer, we observed that the ABOi-H group (≥1:128) experienced significantly higher rates of biliary complications than the ABOc or ABOi-L group (<1:128). Moreover, the duration of BCFS was reduced, and the incidence of DIHBS increased in the ABOi-H group. Anti-ABO antibody titers of ≥1:128 were identified as an independent risk factor for biliary complications after ABOi-LDLT. To determine the criterion for a sufficiently high anti-ABO antibody titer in the initial stage of this study, biliary complications were analyzed by classifying the groups into ABOc, a low titer (<1:64), and high titer (>1:64). No significant difference exists in biliary complications among the three groups (*p* = 0.119) in this setting.

Biliary complications after ABOi LDLT stem primarily from an increased risk of antibody-mediated rejection, which can lead to hepatic necrosis and biliary strictures. When ABO antibodies in the recipient’s bloodstream bind to ABO antigens in donor tissues, an inflammatory cascade with complement activation occurs, involving pro-inflammatory cytokines such as interleukin-6 [13]. This cascade triggers vascular endothelial injury and intravascular thrombus formation within the liver graft. This causes impaired blood circulation in the graft, bile duct ischemia, and hepatic necrosis. These events may result in the development of biliary strictures, cholestasis, and graft loss [12]. Diffuse intrahepatic bile duct complications appear to be more common in ABOi LDLT than in ABOc LDLT, as reported in several studies. This is thought to be related to immunological responses [10].

Graft failure associated with ABO incompatibility can be attributed to hepatic necrosis or DIHBS [14]. The fulminant type of hepatic necrosis markedly decreased after ABOi LDLT because of the introduction of rituximab, which attenuates the production of anti-ABO antibodies by targeting CD20-positive B cells [3]. However, DIHBS remains an unresolved problem of graft failure after ABOi LDLT. Refractory cholangitis after immunological injury to the vascular endothelial cells or bile duct epithelium is an important cause of DIHBS [15], which can eventually lead to graft failure. DIHBS does not necessarily cause death; however, repeated interventions are required for its management, and re-transplantation is the only proven treatment [16].

In this study, no non-anastomotic biliary strictures were observed when the patient was first diagnosed with biliary complications. However, five patients developed DIHBS, constituting 12.8% of the biliary complications. Two of these patients eventually underwent re-transplantation. The progression rate of DIHBS in the ABOi-H group was remarkably higher than that in the ABOc and ABOi-L groups (42.8% vs. 8.3% and 0%, respectively). Among the three patients who experienced DIHBS in the ABOi-H group, one showed an increased postoperative anti-ABO antibody titer of 1:64 (dithiothreitol, 1:512). The titers of the other two patients were 1:16 (1:32) and 1:4 (1:16), respectively. This suggests that DIHBS is not necessarily accompanied by an increase in postoperative anti-ABO antibody titers. 

In this study, duct anastomosis was usually performed using a duct-to-duct anastomosis with an internal stent. Neither the number nor the diameter of bile ducts were correlated with post-LDLT biliary complications. Bile duct reconstruction was performed by skilled surgeons. Therefore, greater consideration was given to immunologic or ischemic factors rather than technical issues, which are crucial in understanding bile duct complications [17,18]. Performing RYHJ for biliary anastomosis is a significant risk factor for the increased incidence of non-anastomotic and anastomotic strictures [8]. However, RYHJ was performed on only 11 patients, and only one of these belonged to the ABOi-L group. Due to the limited representation of ABOi cases compared to ABOc cases, meaningful comparisons with RYHJ were deemed unfeasible in this study. Additionally, the recipient’s female sex was one of the independent risk factors for biliary complications in LDLT (3.307 [1.271–8.604], *p* = 0.014). However, the impact of female sex on biliary complications in LDLT has not been clearly established.

The importance of hepatic artery flow in liver transplantation has always been emphasized. Hepatic artery occlusion is acknowledged for precipitating ischemic damage to the hepatocytes and biliary tract, which elevates the risk of biliary complications [19,20]. Ischemic injury in the biliary system is a crucial cause of biliary necrosis, which causes biliary strictures [21]. Therefore, the evaluation of hepatic artery flow after LDLT should be performed more closely for early detection and treatment, especially for recipients with a high-baseline anti-ABO antibody titer [22]. Portal vein complications that typically arise from thrombosis or stenosis are usually diagnosed using postoperative sonography or follow-up CT. Once portal vein narrowing is observed, portal vein stent placement is required to achieve resolution. This intervention applies physical pressure to adjacent bile ducts, contributing to an increased incidence of biliary complications. However, vascular complications were not noticeable in this study, which is difficult to interpret because of the small cohort size.

This study has certain limitations. This retrospective study, conducted on a small cohort from a single center, has limitations in interpreting the results. In addition, it is difficult to provide a clear reason for the increased incidence of biliary complications and DIHBS in the ABOi-H group because this mechanism is not known in detail. However, considering the substantial proportion of ABOi-LDLT cases included in this study, it is imperative to approach the findings with caution.

## 5. Conclusions

Although a high baseline anti-ABO antibody titer was the most significant risk factor for biliary complications after LDLT, no significant difference was observed between the ABOi-L and ABOc groups. Additionally, recipients’ female sex and hepatic artery complications were significant risk factors for biliary complications in LDLT. It is essential to establish an individualized surveillance strategy for biliary complications in patients undergoing ABOi LDLT, considering their baseline anti-ABO antibody titers. In addition, it is necessary to elucidate how a high anti-ABO titer contributes to the biliary complication mechanism for the appropriate implementation of ABOi LDLT in the future. Finally, considering its higher biliary complication rate and risk factors, it is recommended that ABOi LDLT with a high baseline anti-ABO antibody titer should be performed at a highly experienced center.

## Figures and Tables

**Figure 1 jcm-13-04789-f001:**
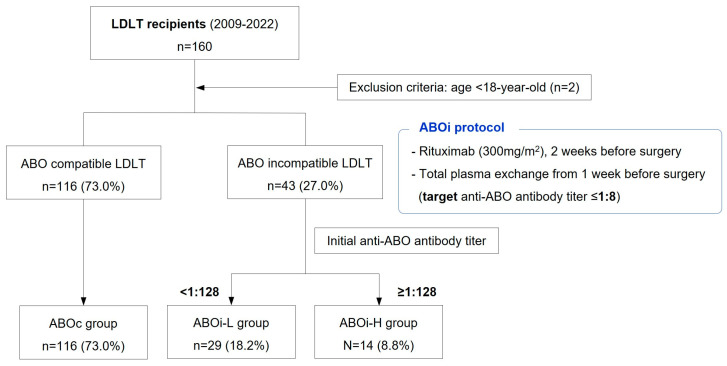
Flowchart of the study population and desensitization protocol. LDLT, living donor liver transplantation; ABOc, ABO-compatible; and ABOi, ABO incompatible.

**Figure 2 jcm-13-04789-f002:**
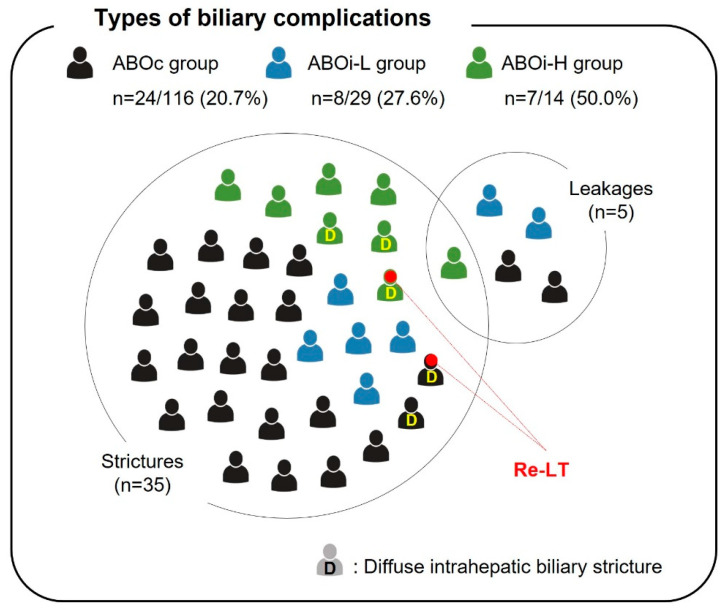
Types of biliary complications among the ABOc, ABOi-L, and ABOi-H groups. The diagram shows the type and distribution of biliary complications in the three groups. The incidence of biliary stricture was significantly higher in the ABOi-H group than in the ABOc and ABOi-L groups (7 [50.0%] vs. 24 [20.7%] and 8 [27.6%], respectively, *p* = 0.041). Two patients with DIHBS underwent re-transplantation due to unresolved biliary strictures (1 in the ABOc group and 1 in the ABOi-H group).

**Figure 3 jcm-13-04789-f003:**
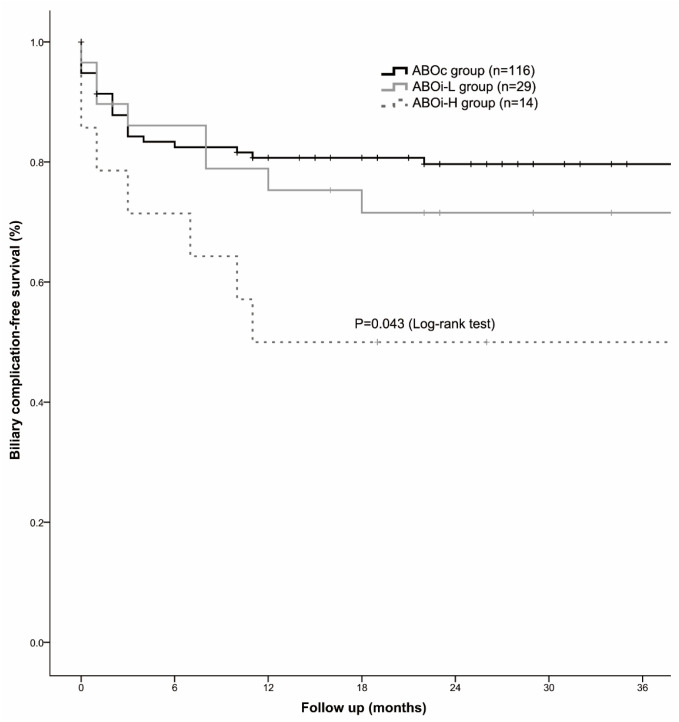
Biliary complication-free survival among the ABOc, ABOi-L, and ABOi-H groups. The ABOi-H group showed a higher incidence of biliary complication-free survival than the other two groups (*p* = 0.043).

**Table 1 jcm-13-04789-t001:** Baseline characteristics of the recipients and donors in the ABOc, ABOi-L, and ABOi-H groups.

	ABOc Group (n = 116)	ABOi-L Group (n = 29)	ABOi-H Group (n = 14)	*p*-Value
Recipients				
Age ^&^	54 (33–73)	55 (34–68)	56 (47–61)	0.862
Sex (male)	85 (73.3%)	16 (55.2%)	11 (78.6%)	0.126
BMI (kg/m^2^) *	23.9 (17.0–40.4)	24.12 (15.5–29.5)	23.73 (20.3–33.2)	0.730
Baseline anti-ABO antibody titer ^¶^		32 (8–64)	256 (128–1024)	<0.001
TPE (number) ^¶^		2 (0–4)	4 (2–11)	<0.001
Diagnosis (multiple)				
HBV	72 (62.1%)	16 (55.2%)	9 (64.3%)	0.766
HCV ^#^	14 (12.1%)	3 (10.3%)	2 (14.3%)	0.922
Alcohol-related cirrhosis	31 (26.7%)	8 (27.6%)	3 (21.4%)	0.902
Autoimmune liver disease ^#^	2 (1.7%)	4 (13.8%)	0 (0.0%)	0.022
Hepatocellular carcinoma	71 (61.2%)	19 (65.5%)	10 (71.4%)	0.718
HTN	21 (18.1%)	9 (31.0%)	3 (21.4%)	0.307
DM	25 (21.6%)	9 (31.0%)	5 (35.7%)	0.339
MELD score at the time of transplantation *	12 (6–40)	11 (6–32)	10 (7–21)	0.288
Child–Pugh Score *	7 (5–14)	7 (5–13)	7 (5–12)	0.632
Platelet (×10^3^) *	70 (18–243)	79 (17–548)	63 (25–144)	0.826
Total bilirubin (mg/dL) *	1.49 (0.29–38.64)	1.54 (0.27–10.49)	1.29 (0.39–5.88)	0.227
PT (INR) *	1.28 (0.92–3.89)	1.23 (0.94–2.37)	1.27 (0.96–1.85)	0.794
Donors				
Age *	30 (16–54)	30 (19–53)	28 (20–55)	0.494
Sex (male)	85 (73.3%)	20 (69.0%)	13 (92.9%)	0.222
BMI (kg/m^2^) *	24.1 (16.7–37.2)	23.8 (19.0–30.3)	22.7 (19.6–34.6)	0.775
ICG-R15 (%) ^&^	9.0 (0.2–18.5)	8.4 (0.3–12.5)	7.4 (1.8–15.3)	0.197
Macrosteatosis (≥20%) ^#^	8 (7.1%)	0 (0.0%)	1 (7.1%)	0.416
Graft type ^#^				
right	93 (80.2%)	17 (58.6%)	11 (78.6%)	0.101
left	18 (15.5%)	11 (37.9%)	3 (21.4%)	
right posterior	5 (4.3%)	1 (3.4%)	0 (0.0%)	
Operation type of donor (laparoscopic)	13 (11.2%)	6 (20.7%)	0 (0.0%)	0.131
GRWR *	0.98 (0.60–1.89)	0.94 (0.70–1.94)	1.05 (0.73–1.32)	0.052

^¶^ Mann–Whitney *U* test, * Kruskal–Wallis test, ^#^ Fisher’s exact test, ^&^ One-way ANOVA; Values are represented as median (range) for continuous data and n (%) for categorical data. BMI, body mass index; TPE, total plasma exchange; HBV, hepatitis B virus; HCV, hepatitis C virus; HTN, hypertension; DM, diabetes mellitus; MELD, model for end-stage liver disease; GRWR, graft-to-recipient weight ratio; PT, prothrombin time; INR, international normalized ratio; and ICG-R15, indocyanine green retention test after 15 min.

**Table 2 jcm-13-04789-t002:** Operative variables and postoperative outcomes among the ABOc, ABOi-L, and ABOi-H groups.

	ABOc Group (n = 116)	ABOi-L Group (n = 29)	ABOi-H Group (n = 14)	*p*-Value
Operative variables				
Operation time (min) *	813 (545–1430)	770 (590–1041)	738 (540–1190)	0.101
Graft’s bile duct features				
Number ^#^				1.000
1	80 (69.0%)	21 (72.4%)	10 (71.4%)	
2	33 (28.4%)	8 (27.6%)	4 (28.6%)	
3	3 (2.6%)	0 (0.0%)	0 (0.0%)	
Ductoplasty	24 (20.7%)	7 (24.1%)	3 (21.4%)	0.921
Maximum size (mm) *	5 (1–13)	4 (2–8)	5 (2–10)	0.990
Minimum size (mm) *	4 (1–13)	4 (2–8)	5 (2–10)	0.909
Multiple hepatic arterial anastomosis ^#^	6 (5.2%)	2 (6.9%)	0 (0.0%)	0.840
Reconstruction type ^#^				0.803
Duct to duct with internal stent	101 (87.1%)	27 (93.1%)	14 (100%)	
Duct to duct without internal stent	7 (6.0%)	1 (3.4%)	0 (0.0%)	
Hepaticojejunostomy	8 (6.9%)	1 (3.4%)	0 (0.0%)	
Duct to duct with internal stent and hepaticojejunostomy	2 (1.7%)	0 (0.0%)	0 (0.0%)	
Warm ischemic time (min) *	5 (0–91)	6 (0–15)	6 (0–41)	0.180
Cold ischemic time (min) *	153 (15–273)	149 (36–228)	142 (98–252)	0.560
Transfused RBC (unit) *	8 (0–117)	10 (0–36)	7 (0–28)	0.936
Postoperative outcomes				
Complication (CD ≥ IIIa)	58 (50.0%)	14 (48.3%)	9 (64.3%)	0.571
Hepatic artery complication ^#^	1 (0.9%)	2 (6.9%)	1 (7.1%)	0.084
Portal vein complication ^#^	5 (4.3%)	2 (6.9%)	0 (0.0%)	0.807
Hepatic vein complication ^#^	12 (10.3%)	2 (6.9%)	2 (14.3%)	0.675
Rejection ^#^	7 (6.0%)	1 (3.4%)	1 (7.1%)	1.000
Biliary complication	24 (20.7%)	8 (27.6%)	7 (50.0%)	0.041
Hospital days ^¶^	20 (1–154)	30 (9–336)	25 (16–50)	0.825
30-day mortality ^#^	1 (0.9%)	0 (0.0%)	0 (0.0%)	1.000
90-day mortality ^#^	3 (2.6%)	1 (3.4%)	0 (0.0%)	1.000

^¶^ Mann–Whitney *U* test, * Kruskal–Wallis test, ^#^ Fisher’s exact test; GRWR, graft-to-recipient weight ratio; CD, Clavien–Dindo classification.

**Table 3 jcm-13-04789-t003:** Univariable and multivariable Cox proportional hazard regression analyses of risk factors for postoperative biliary complications.

	Univariable Analysis		Multivariable Analysis	
	HR (95% CI)	*p*-Value	HR (95% CI)	*p*-Value
Donors				
Age (≥40 years)	0.892 (0.348–2.284)	0.811		
Sex (male)	0.952 (0.464–1.955)	0.894		
BMI (≥30 kg/m^2^)	1.092 (0.263–4.535)	0.904		
Macrosteatosis (≥20%)	0.454 (0.062–3.312)	0.436		
Graft types				
Right graft	Ref.	—		
Left graft	0.304 (0.093–0.991)	0.048	0.430 (1.121–1.537)	0.194
Right posterior graft	2.291 (0.701–7.489)	0.170		
GRWR (<0.8)	1.566 (0.556–4.409)	0.396		
Operation type of donor (laparoscopic)	0.922 (0.328–2.595)	0.878		
Graft bile duct opening (multiple)	2.041 (1.083–3.847)	0.027	1.092 (0.396–3.008)	0.865
Graft ductoplasty	1.112 (0.528–2.343)	0.780		
Minimum diameter of graft bile duct (<4 mm)	0.047 (0.000–362.906)	0.503		
Recipients				
Age (≥60 years)	0.868 (0.399–1.888)	0.721		
Sex (female)	3.088 (1.208–7.897)	0.019	3.307 (1.271–8.604)	0.014
BMI (<18.5 kg/m^2^)	21.490 (0.021–22,167.806)	0.386		
Baseline anti-ABO-Ab titer				
ABO-compatible	Ref.	—		
ABO titer < 128	1.367 (0.613–3.049)	0.445		
ABO titer ≥128	2.815 (1.211–6.542)	0.016	3.943 (1.635–9.506)	0.002
HBV	1.131 (0.587–2.178)	0.712		
HCV	1.471 (0.615–3.519)	0.385		
Alcohol-related cirrhosis	1.004 (0.489–2.064)	0.991		
Autoimmune	0.046 (0.000–42.068)	0.377		
Hepatocellular carcinoma	1.854 (0.903–3.808)	0.093	1.230 (0.563–2.688)	0.604
MELD score (≥20)	0.687 (0.269–1.756)	0.433		
Hepaticojejunostomy	1.808 (0.858–3.810)	0.119		
Warm ischemic time (≥20 min)	1.710 (0.902–3.239)	0.100	1.850 (0.946–3.616)	0.072
Intraoperative transfused RBC (≥20 units)	1.587 (0.838–3.005)	0.156		
Hepatic artery complication	3.119 (1.214–8.018)	0.018	3.505 (1.313–9.354)	0.012
Portal vein complication	1.339 (0.322–5.565)	0.688		
Hepatic vein complication	0.537 (0.191–1.513)	0.240		

HR, hazard ratio; BMI, body mass index; GRWR, graft-to-recipient weight ratio; HBV, hepatitis B virus; HCV, hepatitis C virus; MELD, model for end-stage liver disease; and RBC, red blood cell.

## Data Availability

The data presented in this study are available on request from the corresponding author due to privacy and ethical reasons.

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
