# Peer review of "Impact of Baseline Anti-ABO Antibody Titer on Biliary Complications in ABO-Incompatible Living-Donor Liver Transplantation"

_jcm, 2024, doi:10.3390/jcm13164789_

Round 1

Reviewer 1 Report

Comments and Suggestions for Authors

General comments:

The authors of the paper investigated the impact of the effect of baseline anti-ABO antibody titers on biliary complications after ABO-incompatible (ABOi) living-donor liver transplantation (LDLT) as well as the identification of risk factors for biliary complications following the treatment mentioned above. The paper's subject is interesting, but some issues should be addressed. They are described below.

Specific comments:

Abstract: According to the JCM rules the abstract should be structured into the following parts: background/objectives, methods, results, and conclusions.

Line 25- “a high titer (≥!:128)…” should be corrected to  “the high anti-ABO titer (≥!:128)…”

Line 27- “a high baseline titer…” should be corrected to “a high baseline anti-ABO titer…”

Line 64- “either above or below 1:128…” should be corrected to either equal and above or below 1:128…

Figure 1 and Line 75- there is a difference in the target of anti-ABO antibodies (1:8 vs ≤ 1:8)

≤ 1:8 ≤ 1:8 Figure 1 anti-ABO Ab and anti-ABO antibody used together- it should be unified, abbreviations should be explained in the Figure legend to make it self-explanatory

Line 100- “computed tomography was performed…” should be corrected to computed tomography scanning was performed…

Line 113-  “Statistical analyses were conducted to compare and identify baseline characteristics, operative variables…” should be corrected to Statistical analyses were conducted to compare and identify differences of baseline characteristics, operative variables…

Table 1- the difference in the frequency of autoimmune liver disease should be mentioned as significant next to the other two significant variables of baseline characteristics of patient groups

Line 142- ( 20%) should be corrected to (≥ 20%)

Line 172- p = 0.073 for biliary leakage is not significant

Summary - also the other two significant risk factors for postoperative biliary complications such as recipient female sex (3.307 [1.271–8.604], P = 0.014) and postoperative hepatic artery complications (3.505 [1.313– 9.354], P = 0.012) should be mentioned.

In conclusion: major revision is required

Comments on the Quality of English Language

Minor editing of the English language is required

Author Response

Thank you very much for taking the time to review this manuscript and for your kind comments.

You have provided many valuable suggestions, and we are submitting the revised manuscript based on your comments.

Please see the file uploaded.

Reviewer 2 Report

Comments and Suggestions for Authors

This is a really good and interesting study. However, there are major data workup flaws

1. Why both Fisher's exact test and Chi-square were used? Should not the authors choose one based on sample size? 

2. What were the variables included in the multiVARIABLE (not VARIATE)  analysis?

3. A multivariable linear regression does not provide an OR ( logistic regression provides a OR), but rather a B-coefficient please make necessary modifications.

4. You mentioned ERCP in the diagnostic of biliary strictures could you provide a complication associated with this procedure? this could use some help:  https://doi.org/10.3390/jpm13091356

5. Multivariate analysis should be changed to multivariable analysis in the context of biliary complications. The authors need to specify whether biliary complications are treated as a categorical dichotomous variable (i.e., yes or no) or as an ordinal variable with at least three categories. This clarification will determine the assumptions and appropriate statistical tests to be used. 

Comments on the Quality of English Language

The English language usage in this article is overall quite good. The text is clear and comprehensible, effectively conveying the intended information. However, a few minor revisions are recommended for improved clarity and precision.

Author Response

Thank you very much for taking the time to review this manuscript and for your kind comments.

You have provided many valuable suggestions, and we are submitting the revised manuscript based on your comments.

Response to Reviewers’ Comments [Reviewer 2]

Comment 1: Why both Fisher's exact test and Chi-square were used? Should not the authors choose one based on sample size?

Response 1: Thank you for your comment. We used the Chi-square test when the cohort of the relevant variable for each group was sufficiently large. However, if there were more than 20% of cells with an expected frequency of less than 5, Fisher’s exact test was used to apply an appropriate statistical method for the sample size.

Comment 2: What were the variables included in the multiVARIABLE (not VARIATE) analysis?

Response 2: Thank you for your comment. As pointed out, ‘multivariate’ was changed to ‘multivariable’. We performed a multivariable analysis including variables with a P value of 0.1 or less on the univariable analysis. Those variables are left graft, multiple graft bile duct opening, female sex, ABO-incompatibility with high anti-ABO antibody titer (≥1:128), hepatocellular carcinoma, warm ischemic time of 20 minutes or longer, and hepatic artery complication.

Changes in the text: (Materials and Methods 2.5)

… A multivariablete analysis was performed on factors with P values ≤ 0.1 in the univariablete analysis (left graft, multiple graft bile duct opening, female sex, ABO-incompatibility with high anti-ABO antibody titer [≥1:128], hepatocellular carcinoma, warm ischemic time [≥20 minutes], and hepatic artery complication). …

Comment 3: A multivariable linear regression does not provide an OR (logistic regression provides an OR), but rather a B-coefficient please make necessary modifications.

Response 3: We appreciate you pointing this out. We would like to inform you that the statistical method used in this study, Cox proportional hazard regression, was incorrectly described in the manuscript as linear regression. We apologize for this incorrect information. The statistical method described in Methods, Result, and Table 3 was changed to ‘Cox proportional hazard regression’. Additionally, OR was changed to HR in the revised manuscript.

Changes in the text: (Material and Methods 2.5)

... Cox proportional hazard regression analysis Linear regression analysis was used to determine risk factors for biliary complications.

Changes in the text: (Results 3.4)

… Multivariablete Cox proportional hazard regression linear regression analysis of BCFS revealed that a high baseline anti-ABO antibody titer (≥ 1:128) was an independent risk factor (Hazard ratio [HR]odds ratio, 3.493 [1.635–9.506]; P = 0.002) (Table 3). …

Changes in the text: (Table 3.)

Table 3. Univariablete and multivariablete Cox proportional hazard regression logistic regression analyses of risk factors for postoperative biliary complications

Univariablete analysis

multivariablete analysis

HR OR (95% CI)

P value

HR OR (95% CI)

P Value

Comment 4: You mentioned ERCP in the diagnostic of biliary strictures could you provide a complication associated with this procedure? this could use some help:

https://doi.org/10.3390/jpm13091356

Response 4: Thank you for your comment. Complications after ERCP were reviewed, such as bleeding, pancreatitis, duodenum perforation, and cholangitis. Few patients complained of abdominal discomfort or nausea due to an endoscopic retrograde biliary drainage catheter after the procedure. However, none of the patients included in this study experienced major complications affecting the clinical course. A review of complications after ERCP has been added to the manuscript as below.

Changes in the text: (Result 3.3)

… No significant difference exists among the three groups in the rate of percutaneous transhepatic bile drainage implementation during the treatment period (45.8%, 37.5%, and 42.9%, respectively, P = 1.000,). None of the patients experienced post-ERCP major complications affecting the clinical course, such as bleeding, pancreatitis, or gastrointestinal perforation.

Comment 5: Multivariate analysis should be changed to multivariable analysis in the context of biliary complications. The authors need to specify whether biliary complications are treated as a categorical dichotomous variable (i.e., yes or no) or as an ordinal variable with at least three categories. This clarification will determine the assumptions and appropriate statistical tests to be used.

Response 5: Thank you for pointing this out. All ‘multivariate’ was changed to ‘multivariable’ in the revised manuscript. As mentioned in Response 3, we apologize once again for the confusion caused by the incorrect statistical method description. We used Cox proportional hazard regression in this study. Additionally, the LML (log minus log) graphs of the three groups were running in parallel.

Changes in the text: (same as response 3)

Reviewer 3 Report

Comments and Suggestions for Authors

In the study examining biliary complications in ABO incompatible (ABOi) living-donor liver transplantation (LDLT), the authors present compelling evidence that higher baseline anti-ABO antibody titers are a significant risk factor for adverse outcomes. Additionally, the study highlights the importance of an individualized approach to monitoring and managing patients, particularly those with high antibody titers. The authors make a strong case for meticulous postoperative surveillance in ABOi transplant patients to promptly identify and address potential complications. This work not only contributes significantly to the field by enhancing our understanding of the complexities associated with ABOi LDLT but also provides a roadmap for improving patient care in this challenging clinical area.

The results of the study encourages to continue with AB0 -i transplant. However the authors should mention in the discussion, that this should only be done in specialized centers and the patients should be followed very strict

Author Response

Thank you very much for taking the time to review this manuscript and for your kind comments.

You have provided many valuable suggestions, and we are submitting the revised manuscript based on your comments.

Response to Reviewers’ Comments 

Comment: In the study examining biliary complications in ABO-incompatible (ABOi) living-donor liver transplantation (LDLT), the authors present compelling evidence that higher baseline anti-ABO antibody titers are a significant risk factor for adverse outcomes. Additionally, the study highlights the importance of an individualized approach to monitoring and managing patients, particularly those with high antibody titers. The authors make a strong case for meticulous postoperative surveillance in ABOi transplant patients to promptly identify and address potential complications. This work not only contributes significantly to the field by enhancing our understanding of the complexities associated with ABOi LDLT but also provides a roadmap for improving patient care in this challenging clinical area.

The results of the study encourage to continue with ABOi transplant. However, the authors should mention in the discussion, that this should only be done in specialized centers and the patients should be followed very strict.

Response: Thank you for your comments. We completely agree with your opinion and have added it to the discussion below.

Changes in the text: (Discussion)

… In addition, it is necessary to elucidate how a high anti-ABO titer contributes to the biliary complication mechanism for an appropriate implementation of ABOi LDLT in the future. Finally, considering its higher biliary complication rate and risk factors, it is recommended that ABOi LDLT with a high baseline anti-ABO antibody titer should be performed at a highly experienced center.

Round 2

Reviewer 1 Report

Comments and Suggestions for Authors

The majority of my recommendations have been addressed.

Nevertheless, as I mentioned previously, the other two significant risk factors for postoperative biliary complications such as recipient female sex and postoperative hepatic artery complications should be noted in the conclusions (see abstract). 

Comments on the Quality of English Language

Minor polishing is required.

Author Response

Comment: The majority of my recommendations have been addressed.

Nevertheless, as I mentioned previously, the other two significant risk factors for postoperative biliary complications such as recipient female sex and postoperative hepatic artery complications should be noted in the conclusions (see abstract). 

Response: Thank you very much for pointing this out. We found that the two significant risk factors listed only in the discussion. We revised the conclusion adding the two risk factors (female sex and hepatic artery complications). Thank you again for your kind recommendation.

Changes in the text : (Page 1, Abstract) ... Conclusion. A high baseline anti-ABO antibody titer (≥ 1:128), female sex, and hepatic artery complications were significant risk factors for biliary complications.

Changes in the text : (Page 11, Conclusion) ... Although a high baseline anti-ABO antibody titer was the most significant risk factor for biliary complications after LDLT, no significant difference was observed between the ABOi-L and ABOc groups. Additionally, recipients’ female sex and hepatic artery complications were significant risk factors for biliary complications in LDLT. It is essential to establish an individualized surveillance strategy for biliary complications in patients undergoing ABOi LDLT, considering their baseline anti-ABO antibody titers. ...

Reviewer 2 Report

Comments and Suggestions for Authors

I'm glad authors have made the proper corrections, good luck!

Author Response

Thank you for your valuable comments. We really appreciate your recommendations. Thank you again!